# Case Review: Whole-Exome Sequencing Analyses Identify Carriers of a Known Likely Pathogenic Intronic *BRCA1* Variant in Ovarian Cancer Cases Clinically Negative for Pathogenic *BRCA1* and *BRCA2* Variants

**DOI:** 10.3390/genes13040697

**Published:** 2022-04-15

**Authors:** Wejdan M. Alenezi, Caitlin T. Fierheller, Timothée Revil, Corinne Serruya, Anne-Marie Mes-Masson, William D. Foulkes, Diane Provencher, Zaki El Haffaf, Jiannis Ragoussis, Patricia N. Tonin

**Affiliations:** 1Department of Human Genetics, McGill University, Montreal, QC H3A 0C7, Canada; wagdan.alenizy@mail.mcgill.ca (W.M.A.); caitlin.fierheller@mail.mcgill.ca (C.T.F.); timothee.revil@mcgill.ca (T.R.); william.foulkes@mcgill.ca (W.D.F.); ioannis.ragoussis@mcgill.ca (J.R.); 2Cancer Research Program, Centre for Translational Biology, The Research Institute of McGill University Health Centre, Montreal, QC H4A 3J1, Canada; corinne.serruya@affiliate.mcgill.ca; 3Department of Medical Laboratory Technology, Taibah University, Medina 42353, Saudi Arabia; 4McGill Genome Centre, McGill University, Montreal, QC H3A 0G1, Canada; 5Département de Médecine, Université de Montréal, Montreal, QC H3T 1J4, Canada; anne-marie.mes-masson@umontreal.ca; 6Institut du Cancer de Montréal, Centre de Recherche du Centre Hospitalier de l’Université de Montréal, Montreal, QC H2X 0A9, Canada; diane.provencher.med@ssss.gouv.qc.ca (D.P.); ahmed.zaki.anwar.el.haffaf.med@ssss.gouv.qc.ca (Z.E.H.); 7Lady Davis Institute for Medical Research of the Jewish General Hospital, Montreal, QC H3T 1E2, Canada; 8Department of Medical Genetics, McGill University Health Centre, Montreal, QC H3H 1P3, Canada; 9Department of Medicine, McGill University, Montreal, QC H4A 3J1, Canada; 10Gerald Bronfman Department of Oncology, McGill University, Montreal, QC H3A 1G5, Canada; 11Division of Gynecologic Oncology, Université de Montréal, Montreal, QC H4A 3J1, Canada; 12Service de Médecine Génique, Centre Hospitalier de l’Université de Montréal, Montreal, QC H2X 0A9, Canada

**Keywords:** familial ovarian cancer, whole exome sequencing, *BRCA1*, germline variant, intronic variant, alternative splicing variant

## Abstract

**Background:** Detecting pathogenic intronic variants resulting in aberrant splicing remains a challenge in routine genetic testing. We describe germline whole-exome sequencing (WES) analyses and apply in silico predictive tools of familial ovarian cancer (OC) cases reported clinically negative for pathogenic *BRCA1* and *BRCA2* variants. **Methods:** WES data from 27 familial OC cases reported clinically negative for pathogenic *BRCA1* and *BRCA2* variants and 53 sporadic early-onset OC cases were analyzed for pathogenic variants in *BRCA1* or *BRCA2*. WES data from carriers of pathogenic *BRCA1* or *BRCA2* variants were analyzed for pathogenic variants in 10 other OC predisposing genes. Loss of heterozygosity analysis was performed on tumor DNA from variant carriers. **Results:**
*BRCA1* c.5407-25T>A intronic variant, identified in two affected sisters and one sporadic OC case, is predicted to create a new splice effecting transcription of *BRCA1*. WES data from *BRCA1* c.5407-25T>A carriers showed no evidence of pathogenic variants in other OC predisposing genes. Sequencing the tumor DNA from the variant carrier showed complete loss of the wild-type allele. **Conclusions:** The findings support *BRCA1* c.5407-25T>A as a likely pathogenic variant and highlight the importance of investigating intronic sequences as causal variants in OC families where the involvement of *BRCA1* is highly suggestive.

## 1. Introduction

The major heritable risk factors for ovarian cancer (OC) are pathogenic germline variants in *BRCA1* [1] or *BRCA2* [2]. Women carrying pathogenic variants in either gene are at significantly increased risk for OC from 17% (95% confidence interval [CI]: 11–25%) to 44% (95%CI: 36–53%) by age 80 depending on the gene involved [3], whereas the lifetime risk for OC in the general population is estimated to be 1.2% by age 80 [4,5]. Depending on the population studied, carriers of pathogenic variants in *BRCA1* or *BRCA2* have been identified in 40–85% of OC with a family history of breast cancer (BC) and/or OC (i.e., hereditary breast and ovarian cancer (HBOC) syndrome families) and 10–15% of all epithelial OC [6], regardless of the family history of cancer.

Over 20,000 variants in *BRCA1* and *BRCA2* identified in the context of hereditary BC and/or OC cases have been reported in the literature or in publicly available databases [7,8]. Approximately 15% of all reported variants have been classified as pathogenic or likely pathogenic based on the American College of Medical Genetics and Genomics (ACMG) guidelines [9], and over 90% of these variants were of the nonsense, frameshift or exon-intron splice junction (±1–2 nucleotides from the exon) type, resulting in a purported loss of gene function [7,10]. A pathogenic or likely pathogenic variant classification is mainly under the assumption that such loss-of function (LoF) variants are more likely to result in a premature amino acid termination eliciting nonsense mediated mRNA decay [11,12]. However, other LoF variants located within introns ±3–20 nucleotides from the splice site region that disrupt the normal pattern of mRNA splicing have been described [13,14]. It has been estimated that these splice site variants account for 5% of all pathogenic variants in *BRCA1* and *BRCA2* [15,16].

With advances in sequencing technology, there have been reports of pathogenic variants located deeper in the intronic regions (beyond ±20 nucleotides). These variants introduce new splice sites, affecting gene function [17,18,19]. Depending on the size and complexity of the intronic sequences, these variants can be identified by targeted whole-gene, or whole-genome or -exome sequencing (WGS or WES) methods [19]. The contribution of these variants in *BRCA1* or *BRCA2* to hereditary BC and OC is unknown due to the paucity of studies [20,21,22]. In vitro studies demonstrating the biological impact on splicing [15,23] have led to the reclassification of such variants of unknown clinical significance (VUS) to either pathogenic/likely pathogenic or benign/likely benign [24].

Here, we describe sequencing results for *BRCA1* and *BRCA2* derived from the application of WES and bioinformatic analyses in the re-evaluation of OC cases reported negative for pathogenic variants in these genes by clinical testing. We report the identification of an intronic variant in *BRCA1* (NM_007294.4): c.5407-25T>A harbored by sisters affected with OC. We also report the analysis of this variant in WES data from early-onset OC cases not selected for family history of cancer and OC cases from HBOC families, loss of heterozygosity (LOH) analyses of the *BRCA1* locus in OC tumor DNA and WES analyses of peripheral blood lymphocyte (PBL) DNA of 10 other OC predisposing genes from variant carriers. We relate our observations to independent findings of all *BRCA1* intronic variants that were identified in the context of HBC and HBOC that were curated from ClinVar, a resource that aggregates information about relationships among variation and human health [8], and from a review of the literature.

## 2. Materials and Methods

### 2.1. Study Participants

All study participants were selected from biobanks available through adult hereditary cancer clinics in Quebec and/or the Banque de tissus et données of the Réseau de recherche sur le cancer of the Fond de recherche du Québec–Santé (RRCancer biobank) (rrcancer.ca), who had been recruited for research to biobanks in accordance with ethical guidelines of their respective Institutional Research Ethics Boards. All cases and information about their cancer family history, histopathology, tumor grade, disease stage and/or age of diagnoses was anonymized prior to being provided. This project was conducted with approval and in accordance with the guidelines of The McGill University Health Centre Research Ethics Board (MP-37-2019-4783).

The 27 OC cases investigated for the re-evaluation of *BRCA1* and *BRCA2* included cancers of the ovary or fallopian tubes, or primary peritoneal carcinomas. They are part of 22 OC families defined by having at least two OC cases within first-, second- or third-degree relatives and were selected from the biobanks for this study as they had been previously tested and found negative for pathogenic variants in these genes in medical genetics settings. The characteristics of 16 OC cases from 14 families of French Canadian (FC) ancestry of Quebec have been described previously [25,26,27]; and the remaining 11 OC cases from eight families self-reported having European ancestries. The average age at diagnosis for 25 of 27 OC cases is 55 (median: 57 and range: 25–74) years, as this data was not provided for two cases. These cases comprised serous, high-grade serous carcinoma (HGSC), high-grade endometrioid carcinoma and carcinomas of mixed OC subtypes.

The study participants investigated for the identified *BRCA1* variant have been described previously [25,26,27]: 53 sporadic early-onset OC cases, not selected for family history of cancer who were diagnosed before the age of 50 years and tested negative for pathogenic *BRCA1* or *BRCA2* pathogenic variants; and 24 OC cases from 22 HBOC syndrome families who previously tested positive for pathogenic *BRCA1* or *BRCA2* variants. All cases self-reported FC ancestry of Quebec and had undergone medical genetic testing in adult hereditary cancer clinics in Quebec, as described elsewhere [27].

### 2.2. WES Analysis of BRCA1 and BRCA2 Loci

*BRCA1* (NM_007294.4) and *BRCA2* (NM_000059.4) loci were investigated for pathogenic variants in PBL DNA from 27 OC cases from 22 families. These cases were subjected to WES and a customized bioinformatics pipeline for germline variant calling at the McGill Genome Center, as previously described [27]. NimbleGen SeqCap^®^ EZ Exome Kit v3.0 (Roche, USA) was used to capture 64 mega base pairs of coding, non-coding and flanking intronic regions of up to 100 base pairs, where the average coverage of the coding region was 100×, and that of the flanking intronic region was 60× [28,29]. Then, the annotated variant call format (VCF) files were subjected to additional filtering and prioritizing criteria as follows. We filtered WES data for all rare variants in *BRCA1* and *BRCA2* with a minor allele frequency (MAF) ≤ 0.005 in the non-cancer general population using Genome Aggregation Database (gnomAD) v2.1.1 (gnomad.broadinstitute.org) [30,31]. This database reports on WES and WGS data from different populations, including MAF of rare pathogenic variants found in less than 1 in 10,000 individuals in the general population [9]. Loci with total coverage of <10 and/or alternative variant frequency of <0.2 or >0.8 were filtered out in order to reduce the rate of false positive variants [32]. Candidate variants were verified manually using the Integrative Genomics Viewer (IGV) v2.8 [33].

The identified candidate variant was verified in the carrier’s PBL DNA using bidirectional Sanger sequencing of amplified PCR products using customized primers (forward: 5′-ACAGTAGGACCTCATGTCTACA-3′; and reverse: 5′-ATGGAAGCCATTGTCCTCTG-3′) at the McGill Genome Center, as previously described [27]. Sequencing chromatograms were then visually inspected for the variant using 4Peaks v1.8 (nucleobytes.com/4peaks/) (The Netherlands Cancer institute, Amsterdam, The Netherlands).

### 2.3. Databases and In Silico Tools for the Evaluation of BRCA1 and BRCA2 Variants

Variants identified in *BRCA1* and *BRCA2* were assessed for their conservation and deleteriousness at RNA and protein levels using different in silico tools, which were selected based on their best predictive performance as previously described [19,34,35,36]. Variants were also evaluated for their clinical classification as pathogenic or likely pathogenic in the context of cancer using the ClinVar database [8], ACMG guidelines [9] and BRCA exchange database [7].

### 2.4. Surveying Carrier Frequencies in Other in-House Sequencing Data of OC Cases

We surveyed our available in-house WES data that had previously been generated from PBL DNA from different study groups to identify additional carriers of the identified *BRCA1* variant and verify *BRCA1* and *BRCA2* status. This group consisted of 53 sporadic early-onset OC cases who were reported to have tested negative in clinical settings for pathogenic variants in *BRCA1* and *BRCA2* and 24 familial OC cases from 22 HBOC syndrome families who had tested positive for pathogenic variants in *BRCA1* and *BRCA2*. All cases were subjected to the same WES capture kits and bioinformatics pipeline for germline variant calling applied for the 27 cases, as described above.

### 2.5. Profiling Tumor DNA from BRCA1 c.5407-25T>A Variant Carriers

LOH analysis was performed by Sanger sequencing of OC tumor DNA from *BRCA1* c.5407-25T>A variant carriers using customized primers as described above. Extracted DNA from fresh-frozen (FF) or histopathological sections from formalin-fixed paraffin-embedded (FFPE) tumor tissues were provided (RRCancer biobank) for DNA extraction based on the manufacturer’s instructions (Promega, Canada). Sequencing chromatograms were then visually inspected for loss of the wild-type allele, as above.

### 2.6. Characterization of BRCA1 c.5407-25T>A Carriers for Co-Occurring Pathogenic Variants in Other Known OC Risk Genes

We extracted all variants in the following 10 known OC risk genes [37] from WES data from the three *BRCA1* c.5407-25T>A variant carriers and all of the remaining 25 familial OC cases: *MLH1* (NM_000249.4), *MSH2* (NM_000251.3), *MSH6* (NM_000179.3), *PMS2* (NM_000535.7), *BRIP1* (NM_032043.3), *RAD51C* (NM_058216.3), *RAD51D* (NM_001142571.2), *STK11* (NM_000455.5), *PALB2* (NM_024675.4) and *ATM* (NM_000051.4) selected based on the National Comprehensive Cancer Network (NCCN) Clinical Practice in Oncology Guidelines 2020 (Version 2.2021)—Genetic/Familial High-Risk Assessment: Breast, Ovarian and Pancreatic [38].

WES data were filtered for rare variants with MAF ≤ 0.005 based on the non-cancer general population using gnomAD v2.1.1 (gnomad.broadinstitute.org) [30,31]. The missense and intronic variants on this list were further prioritized as predicted to be damaging or affect splicing using in silico tools selected based on their best predictive performance [19,34,35,36]. Rare variants from this list were then prioritized for further review based on their clinical classification as pathogenic or likely pathogenic in the context of cancer using the ClinVar database [8] and ACMG guidelines [9].

## 3. Results

### 3.1. WES and Bioinformatics Analysis Identified BRCA1 c.5407-25T>A as a Candidate Pathogenic Variant

By applying our bioinformatics pipeline and filtering criteria on WES data from 27 OC cases from 22 families, we identified a total of four *BRCA1* and six *BRCA2* variants (Appendix A). The variants were identified in six cases from five families, meaning some OC cases harbored more than one variant, and as described further below, there was one family of siblings harboring an identical *BRCA1* variant.

From the list of seven exonic variants identified by our methods, all but *BRCA2* c.4570T>G; p.Phe1524Val were not predicted to be damaging by all seven selected in silico tools (see Appendix A). Furthermore, all these coding variants have been classified as benign by ClinVar and/or benign or likely benign by ACMG guidelines in the context of hereditary BC and/or OC and reported as benign in the BRCA exchange database [7]. These observations are consistent with medical genetic reports for cases harboring these variants, as commercial testing should have detected these exonic variants if regions were adequately covered.

The three intronic *BRCA1* variants identified by our methods are interesting as they may not have been detectable by commercial testing. The allele frequencies of *BRCA1* c.134+1335del and c.4358-722del are unknown, as neither variant was identified in gnomAD. In contrast, in gnomAD *BRCA1* c.5407-25T>A is infrequent in the non-cancer general population having a MAF of 7.46 × 10^−6^, and a carrier frequency of two out of 134,138 individuals from the non-Finnish European population, a population of ancestral origin closest to the ancestry of our cancer families. As these intronic variants are located beyond ±20 nucleotides from splice sites, none of the prediction scores for affecting splicing were generated by the in silico tools Maximum Entropy Estimates of Splice junction strengths v2.0 (MaxEntScan v2.0) [39], Human Splicing Finder v3.1 (HSF v3.1) [40] and two Database Splicing Consensus Single Nucleotide Variant (dbscSNV) in silico tools: AdaBoost v4.0 (ADA v4.0) and Random Forest v4.0 (RF v4.0) [41] (Appendix A). Therefore, we used SpliceAI, a relatively new in silico tool based on a deep learning and pre-mRNA transcript sequencing database, which generates different scores between 0 and 1 that can be interpreted as the probability of the variant affecting splicing by the loss or gain of a splice acceptor or a splice donor site [19]. SpliceAI predicted *BRCA1* c.5407-25T>A may result in splice acceptor loss (delta score for acceptor loss = 0.41) (Appendix A), suggesting that it might exert a deleterious effect on the transcription of *BRCA1*. Moreover, the locus of *BRCA1* c.5407-25T>A is predicted to be conserved by in silico tools, supporting its biological importance [42]. In contrast, the other two intronic variants were not predicted to affect splicing of *BRCA1* by SpliceAI (Appendix A). Using IGV, a manual inspection of the sequencing reads for *BRCA1* c.134+1335del and c.4358-722del revealed that they are located within repetitive regions deep within introns 3 and 12 of *BRCA1*, respectively, suggesting that sequencing data could be due to technical artifacts [43]. Indeed, association with risk is questionable as both intronic variants have been classified as benign in ClinVar and by ACMG guidelines, though they had not yet been reviewed in the BRCA exchange database (Appendix A).

*BRCA1* c.5407-25T>A is located at base pair 25 of intron 21 upstream from the start of exon 22 based on the transcript NM_007294.4 (Figure 1A), or of intron 22 upstream from the start of exon 23 based on the canonical transcript NM_007300.4, a transcript containing exon 4, which was missed due to a historical misannotation of an additional exon 4 in *BRCA1* [44]. In this report, we have annotated our variants using the *BRCA1* transcript (NM_007294.4), as it is the commonly used in the clinical genetic setting. Interestingly, this variant was identified in two affected sisters with cancer (Figure 1B), and a manual review of their sequencing files using IGV (Figure 1A) shows an average coverage of 60× by our WES capture kit as has been demonstrated by gnomAD v2.1 WES data (gnomad.broadinstitute.org/gene/ENSG00000012048?dataset=gnomad_r2_1 accessed on 5 February 2021). The variant was verified by bidirectional Sanger sequencing of PBL DNA (Figure 1C) from both of our carriers.

### 3.2. WES Analyses Identified Another OC Case Harboring BRCA1 c.5407-25T>A

To determine whether *BRCA1* c.5407-25T>A occurs in other OC cases from our study groups, we reviewed similarly derived WES data sets from familial and sporadic OC cases. This variant was not identified in WES data from 24 OC cases from HBOC families harboring pathogenic *BRCA1* or *BRCA2* variants. In contrast, a carrier was identified among 53 OC cases who developed HGSC before the age of 50 years. Interestingly, this case (PT0198) had previously been reported as negative for pathogenic variants in *BRCA1* or *BRCA2* as well as in *MLH1*, *MSH2*, *MSH6* and *PMS2*. The sporadic cancer case harboring the *BRCA1* variant was diagnosed with HGSC at the age of 45 years, and the variant was verified by bidirectional Sanger sequencing of their PBL DNA (Figure 1D).

### 3.3. WES Analyses of BRCA1 c.5407-25T>A Carriers Suggest That They Are Unlikely to Harbor Pathogenic Variants in the Other Known OC Risk Genes

To further support the role of *BRCA1* c.5407-25T>A in OC risk, we extended our WES and bioinformatic analyses to include an investigation of other known OC risk genes [38] in the familial OC (PT0141 and PT0140, see Figure 1B), and early onset OC (PT0198) cases harboring this *BRCA1* allele. A review of WES data for pathogenic variants in *MLH1*, *MSH2*, *MSH6*, *PMS2, BRIP1*, *RAD51C*, *RAD51D*, *PALB2*, *ATM* or *STK11* identified a carrier of *MSH6* (NM_000179.2): c.-18G>T in PT0140 from family F1612. This variant is classified as benign or likely benign by six submissions in ClinVar (Accession number VCV000089159.7) and likely benign by ACMG guidelines. A similar analysis of the WES data from PT0198 did not identify any pathogenic variants in these genes.

A similar analysis of WES data in remaining 25 familial OC cases did not identify any variants classified as pathogenic or likely pathogenic in any of the 10 OC risk genes. Indeed the only rare variant identified, a missense variant in *BRIP1* (NM_032043.3): c.2220G>T; p.Gln740His harbored in two sisters (PT0204 and PT0217) from family F1608 was not predicted to be damaging by five out of the seven selected in silico tools. Furthermore, this variant has been classified as likely benign by 6 submissions and VUS by 15 submissions in ClinVar (VCV000133752.33) and VUS by ACMG guidelines in the context of hereditary BC and/or OC.

However, our analyses of the sporadic early-onset OC case (PT0198), which included a thorough investigation of *BRCA1* and *BRCA2* loci, revealed that they also harbored *BRCA2* c.1938C>T; p.Ser646Ser, a variant also found in one of the familial OC carrier cases. As discussed above, this synonymous variant in *BRCA2* is classified as benign (Appendix A). Thus, our findings suggest that *BRCA1* c.5407-25T>A carriers are unlikely to harbor pathogenic, likely pathogenic or VUS in the other known OC predisposing genes based on NCCN guidelines for clinical practice in oncology [38].

### 3.4. LOH Analysis of the Tumor DNA from BRCA1 c.5407-25T>A Carrier Revealed Loss of the Wild-Type Allele

We performed LOH analysis of tumor DNA from PT0198, the only available OC tumor DNA from *BRCA1* c.5407-25T>A carriers. DNA sequencing analyses suggested loss of the wild-type allele had occurred in the development of OC in this case (Figure 1D). This finding is consistent with *BRCA1* c.5407-25T>A, playing a role in OC risk, as has been shown with LOH analyses of OC tumor DNA from carriers of *BRCA1* pathogenic variants [45].

## 4. Discussion

Our WES and bioinformatics analyses of 27 familial OC cases who had undergone medical genetic testing and who were reported as negative for pathogenic variants in *BRCA1* and *BRCA2* identified two sisters harboring *BRCA1* c.5407-25T>A. Further investigation of WES data from additional OC cases identified another carrier of this rare *BRCA1* variant among the cases who were also previously reported as having tested negative for pathogenic variants in *BRCA1* or *BRCA2*. As our WES data captured some intronic sequencing data, it is possible that this variant was not detected by medical genetic testing efforts due to commercial testing protocols that were applied.

Our application of SpliceAI, a new tool capable of predicting splice sites up to 10 kilobase pairs from the exon-intron junctions [19], predicted that this intronic variant may result in a splice acceptor loss. This suggests that the nucleotide substitution at −25 from exon 22 along with the adjacent nucleotide at −24 created a new splice acceptor site, potentially resulting in the loss of the entire or part of exon 22 of *BRCA1* (see Figure 1E). The sensitivity of predicting aberrant splicing effects is estimated to be at least 70% for intronic variants between ±20–50 base pairs from exon–intron junctions [19]. Intronic regions containing sequences of potential exonic characteristics are referred to as pseudoexons or exons where a single substitution or small deletion or insertion may create new splice sites, such that these pseudoexons would be recognized by splicing machinery and result in abnormal patterns of splicing [17,46]. The application of SpliceAI has been used recently in different disorders [47].

*BRCA1* c.5407-25T>A was identified in siblings both having OC (one diagnosed with a HGSC of the ovary and the other with primary peritoneal carcinomatosis), cancer phenotypes consistent with harboring a *BRCA1* or *BRCA2* pathogenic variant [48,49]. Indeed, applying the Manchester Scoring System revealed a probability greater than 10% of either sibling harboring a pathogenic variant in *BRCA1* (score = 23) or *BRCA2* (score = 15) [50,51,52]. Interestingly, we also found this variant in one of the early-onset OC cases who developed HGSC before the age of 50 years, which is consistent with observations that the average age of diagnosis of HGSC is less than 60 years of age in carriers of pathogenic *BRCA1* variants [3]. Moreover, our genetic analysis of OC tumor DNA from this carrier revealed the loss of the wild-type allele and retention of the *BRCA1* variant allele. This observation is consistent with *BRCA1* c.5407-25T>A, playing a role in OC risk, as has been shown with LOH analyses of OC tumor DNA from those harboring loss-of-function pathogenic variants in *BRCA1* [45]. *BRCA1* c.5407-25T>A had initially been classified as a VUS in ClinVar based on two submissions of its identification in the context of HBC or HBOC (Appendix A) and five independent studies published prior to 2020 that also described its identification in this hereditary cancer context (Table 1). The *BRCA1* variant was identified via different detection platforms such as protein truncation test, single-strand conformational polymorphism analysis [53] or multiplex ligation-dependent probe analysis. In 2020, during the course of our investigation, Høberg-Vetti et al. reported the identification of *BRCA1* c.5407-25T>A in BC (*n* = 12) and OC (*n* = 8) cases, which also included a case of peritoneal carcinomatosis, in 20 cancer families with *BRCA1* Manchester scores ranging from 3 to 30 [54]. Indeed, the authors mentioned that they had identified this variant as early as 2006 and had evidence from one case of an effect on *BRCA1* transcript [55], though the results were not published [54], highlighting the complexity of interpreting intronic variants. The prevalence of this variant reported in the Norwegian study groups (see Table 1) suggests the possibility of common ancestry for carriers, as has been shown with specific pathogenic variants in defined populations from our study of French Canadians and described in other studies [36]. Høberg-Vetti et al. also provided evidence that *BRCA1* c.5407-25T>A creates a new splice site, resulting in the skipping of exon 22, based on a deletion of 61 nucleotides deduced from sequencing the corresponding aberrant size transcript [54]. This could affect protein function as it would result in the partial deletion of the BRCA1 Carboxy-Terminus (BRCT) domain (Figure 2), and thereby affect the binding of several proteins such as BRIP1, RAP80 and CtIP, which mediate the recruitment or stability of BRCA1 [56]. However, this group also demonstrated that the shift in the reading frame, which introduces a premature termination codon after 11 amino acids *BRCA1* p.Gly1803GlnfsTer11, likely triggers nonsense-mediated mRNA decay [54]. As RNA is not available from our *BRCA1* c.5407-25T>A carriers, we are unable to replicate these findings. Although more research on OC risk associated with *BRCA1* c.5407-25T>A is required, collectively, these observations are supportive of the ClinVar classification of likely pathogenic rather than VUS.

The frequency of pathogenic *BRCA1* intronic variants is currently unknown but likely underreported due to complex methods used to identify them and assess their biological and clinical impact. While researching independent evidence for the pathogenicity of *BRCA1* c.5407-25T>A, we surveyed the ClinVar database (Figure 2A and Appendix A) for rare, intronic *BRCA1* variants located beyond ±20 nucleotides, rationalizing that this resource would report variants with biologically meaningful associations with cancer risk. A literature review revealed that intronic variants are being identified using a variety of DNA and RNA sequencing technologies, including reverse-transcribed- [60,61], long-range- [62] and multiplex- [63] PCR-based assays, some of which aim to identify variants within intronic regions as large as 10 kilobase pairs [15]. Recently, next-generation sequencing technologies involving RNA [64] or whole genome [18,65] sequencing have been applied. The biological impact of intronic variants can be difficult to discern but usually involve in cellulo assays of genetically engineered cell lines sometimes derived from carriers or minigene constructs [66,67]. Our survey of the ClinVar database revealed that 0.3% (35/11,366) of all reported *BRCA1* variants were rare intronic variants that met our criteria (see Figure 2A), where 46% (16/35) were identified between ±20 and ±50 nucleotides and the remaining beyond ±50 nucleotides from exonic-intronic junctions. In total, 2 of the 35 intronic variants were listed as having conflicting interpretation, the variant of our interest c.5407-25T>A (VCV000371817) as VUS or likely pathogenic and c.5153-26A>G (VCV000125786) as VUS or likely benign; and the remaining 33 variants were classified as VUS. Only 3 of 35 *BRCA1* intronic variants listed in ClinVar were identified in the gnomAD database, and this included our variant of interest, *BRCA1* c.5407-25T>A (Appendix A).

From a literature review, we curated a list of 223 original research studies or case reports about rare intronic variants in *BRCA1* (Figure 2B and Appendix A). This list included 32 reports describing 80 such variants. Of these intronic variants, 21% (17/80) were identified between ±20 and ±50 nucleotides and the remaining beyond ±50 nucleotides from exonic–intronic junctions. Only 2 of 80 variants had been classified as pathogenic c.4185+4105C>T (VCV000632611.2) or likely pathogenic c.5407-25T>A, our variant of interest, and the remaining as VUS. Only 4 of the 80 variants were identified in the ClinVar database, and all had been classified as VUS based on ACMG guidelines [9].

Some studies have argued that the majority of deep intronic variants are unlikely to be associated with cancer risk [68]. Interestingly, the frequency of intronic variants predicted by in silico tools to affect splicing is comparable to those predicted to effect bona fide splice site regions [19,69]. We applied SpliceAI [19] to predict the effect in splicing of *BRCA1* to the *BRCA1* intronic variants identified in ClinVar and in our literature search. Unlike our findings with *BRCA1* c.5407-25T>A, SpliceAI predicted that the majority of curated intronic variants would not affect splicing, though the accuracy of this in silico tool did not reach 95% for all applications (see Appendix A). New in silico tools have been developed to predict the splicing impact by these intronic variants using different mathematical models such as CADD-Splice [69] and SQUIRLS [70]. None of these tools have been tested yet on hereditary cancer syndromes. Earlier-developed in silico predictive models include MaxEntScan [39], HSF [40] or both dbscSNV tools [41]: ADA or RF were designed to predict variants within the splice regions. These tools have been tested on different datasets including sequencing data from hereditary BC and OC cases [71]. However, these in silico tools are limited to predicting events that occur within splice regions. Although in cellulo assays would provide supportive evidence for biological impact predicted by bioinformatic tools, the causality of an intronic variant identified in an established highly penetrant cancer predisposing gene such as *BRCA1* in conferring risk to cancer remains a challenge.

Clinical testing for pathogenic variants in *BRCA1* and *BRCA2* has been established in clinical settings as it has been proven to improve cancer risk assessment and management of carriers [37]. OC cases harboring pathogenic variants in these genes are also offered targeted chemotherapies based on poly(ADP-ribose) polymerase (PARP) inhibitors as part of the their standard-of-care treatment regimens, as carriers have shown improvement in overall outcome [72,73]. As sequencing information is gathered from OC patients undergoing different treatment regimens, it will be interesting to investigate the response to PARP inhibitors in carriers of deep intronic variants, particularly those predicted to affect splicing and impact gene function by in silico analyses or by in cellulo assays. Our survey of ClinVar and the literature identified 105 rare intronic *BRCA1* variants that are classified as VUS for further confirmation of their pathogenicity. Although our study may have been limited by whole-exome sequencing, our report highlights the importance of the comprehensive sequencing of the entirety of *BRCA1* and *BRCA2* to capture all possible pathogenic variants in individuals at risk for hereditary OC and BC.

## 5. Conclusions

Using our WES and bioinformatics analyses, we were able to identify an intronic variant in *BRCA1* in one OC family who had tested negative for pathogenic variants in *BRCA1* or *BRCA2* by commercial testing. We also identified this variant in another OC case diagnosed at an early age and showed loss of the wild-type allele in the tumor DNA using LOH analysis. A splice predictor algorithm suggests that it exerts aberrant splicing affecting gene function. Our findings support *BRCA1* c.5407-25T>A as a likely pathogenic variant and highlights the importance of investigating any intronic variants as causal variants in OC families where the involvement of *BRCA1* is highly suggestive.

## Figures and Tables

**Figure 1 genes-13-00697-f001:**
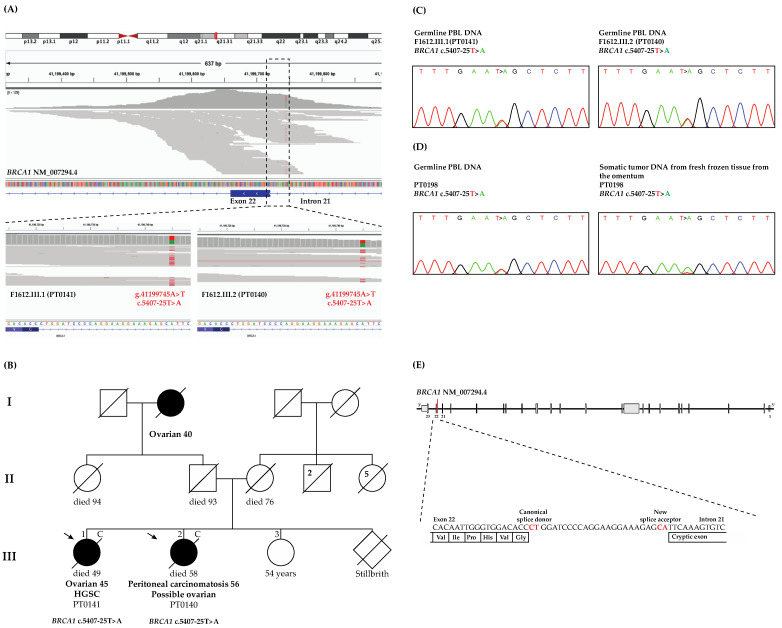
Identification of *BRCA1* c.5407-25T>A, a likely pathogenic intronic variant. (**A**) Integrative Genomics Viewer (IGV) v2.8 sequencing data showing coverage of exon 23 of *BRCA1* and flanking intronic regions beyond ±20 nucleotides of location of *BRCA1* c.5407-25T>A in variant carriers; (**B**) pedigree of *BRCA1* c.5407-25T>A carrier family (F1612) indicating confirmed (“C”) cases of bilateral high-grade serous carcinoma (HGSC) (PT0141) and primary peritoneal carcinomatosis with possible ovarian origin (PT0140) by pathology reports; (**C**) sequencing chromatogram of peripheral blood lymphocytes (PBL) DNA verifying heterozygous carrier status of both sisters; and (**D**) sequencing chromatograms from a sporadic early-onset OC case (PT0198) verifying heterozygous *BRCA1* variant carrier status in PBL DNA and loss of the wild-type allele in OC tumor DNA; (**E**) Schema illustrating the location of the intronic *BRCA1* variant creating a new splice site as predicted by SpliceAI.

**Figure 2 genes-13-00697-f002:**
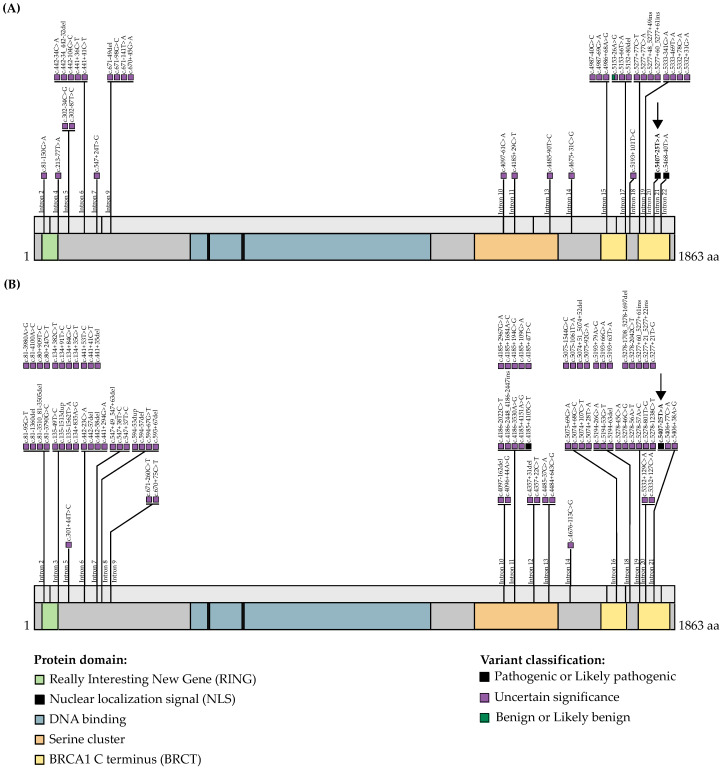
Annotated curated intronic *BRCA1* variants. The *BRCA1* transcript NM_007294.4 (NCBI Reference Sequence (RefSeq) database (ncbi.nlm.nih.gov/nuccore/NM_007294.4), indicating protein encoded domains was annotated with *BRCA1* c.5407-25T>A (indicated with an arrow) and intronic variants classified as pathogenic, likely pathogenic or unknown significance based on ClinVar and/or American College of Medical Genetics and Genomics guidelines were (**A**) submitted to ClinVar (ncbi.nlm.nih.gov/clinvar/ accessed on 25 January 2022) (see Appendix A) or (**B**) reported in the literature (pubmed.ncbi.nlm.nih.gov accessed on 25 January 2022) (see Appendix A). Variants were selected based on: intronic location beyond ±20 nucleotides from splice sites; rarity (minor allele frequency ≤0.005); classification as pathogenic, likely pathogenic or variants of unknown significance (VUS) in *BRCA1* in the context of hereditary breast and/or ovarian cancer Search terms for PubMed (as in October 2021) articles included: (“brca1 s”[All Fields] OR “genes, brca1”[MeSH Terms] OR (“genes”[All Fields] AND “brca1”[All Fields]) OR “brca1 genes”[All Fields] OR “brca1”[All Fields]) AND (“intron s”[All Fields] OR “intron”[All Fields] OR “intronically”[All Fields] OR “intronization”[All Fields] OR “introns”[MeSH Terms] OR “introns”[All Fields] OR “intron”[All Fields] OR “intronic”[All Fields]) AND (“variant”[All Fields] OR “variant s”[All Fields] OR “variants”[All Fields]). Intergenic, 3′UTR and 5′UTR variants and large chromosomal rearrangements were excluded.

**Table 1 genes-13-00697-t001:** Features of *BRCA1* c.5407-25T>A carriers from independent reports.

Year Reported	Population ^1^	Number of Carriers per Study Group	Cancer Type in Carriers	Study Group Investigated ^2^	Reference
2003	Germany	1/90	Breast	Early-onset cases not selected for family history of cancer	[53]
2014	Greece	1/473	Breast	HBC and HBOC families	[57]
2016	Norway	2/893	Breast	Cases not selected for family history of cancer	[55]
2018	Norway	9/669	Breast	HBC and HBOC families	[58]
2019	Norway	8/1914	Breast or ovarian	Sporadic cases and families	[59]
2020	Norway, France, United States of America	20	Breast or ovarian	Selected HBC and HBOC families	[54]
2022	French Canadian, Ashkenazi Jewish, Austria, United Kingdom, Germany, Italy	2/27	Ovarian	Families with at least two OC case within first-, second- or third-degree relatives	This report
2022	French Canadian	1/53	Ovarian	Sporadic OC case with early onset of the disease not selected for family history of cancer	This report

^1^ Geographic origin of population or self-reported as French Canadian from Quebec. ^2^ Cases investigated include hereditary breast cancer (HBC) and hereditary breast and ovarian cancer (HBOC) families.

## Data Availability

Not applicable.

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
