# Peer review of "Case Review: Whole-Exome Sequencing Analyses Identify Carriers of a Known Likely Pathogenic Intronic *BRCA1* Variant in Ovarian Cancer Cases Clinically Negative for Pathogenic *BRCA1* and *BRCA2* Variants"

_genes, 2022, doi:10.3390/genes13040697_

Round 1

Reviewer 1 Report

In the paper by Alenezi et al., the authors examine 27 familial ovarian cancer (OC) cases and 53 sporadic early-onset OC cases all reported negative for BRCA1/BRCA2 pathogenic variants by WES analysis. The authors analyze for pathogenic variants in BRCA1/BRCA2 and report the identification of a known likely pathogenic intronic BRCA1 variant (c.5470-25T>A) in 1 familial OC case (2 sisters) and one sporadic early-onset case. The variant is a recurrent variant in Norway and results in partial skipping of exon 23. The BRCA1 c.5470-25T>A cases were also examined for variants in 10 additional OC predisposing genes and harbour no pathogenic variants in these genes. Finally, LOH analysis was performed on tumor DNA from one BRCA1 c.5470-25T>A carrier and revealed loss of the wild-type allele. The authors conclude that their findings support the BRCA1 c.5470-25T>A variant as likely pathogenic.

Taken together, the manuscript is well-written, and the data is clearly presented.

Major points

  • It is strange that the authors use the alternative BRCA1 transcript NM_007300.4. Please use the most used transcript NM_007294.4 and the nomenclature c.5407-25T>A.

  • Please include pedigree data from the sporadic OC BRCA1 c.5407-25T>A carrier in Figure 1.

  • Since the BRCA1 finding is of limited novelty, I would suggest that the authors include findings from the 10 other OC predisposing genes in the 27 familial OC cases as well as the 53 sporadic cases in the manuscript.

  • In the introduction (line 82-84) the authors mention that WES analysis can identify intronic sequences. Moreover, the authors mention deep intronic sequences several times in the manuscript. I would not use WES for deep intronic analysis. Please discuss in more detail.

  • In the results section (line 215-220), the authors mention that none of the prediction programmes predict that the c.5407-25T>A variants affects splicing. However, MaxEntScan predicts the generation of weak splice acceptor site. This should be mentioned in the manuscript.

Minor points

  • Regarding the title suggest changing “clinically negative” to “previously tested negative”.

  • In the abstract, the author mention that LOH analysis is performed on tumor DNA from variant carriers. It is performed on tumor DNA from one carrier. Moreover, the authors state that the c.5470-25T>A variant is predicted to create a new cryptic splice site. However, RNA analysis has shown that the variants indeed affects splicing (PMID: 32203205), so I would suggest that the authors mention this instead of predicted data.

  • The BRCA1 c.5407-25T>A variant does not activate a cryptic splice site but introduce a de novo splice acceptor site.

  • The manuscript lacks some references (my version only has 37 references)

Author Response

Please view the attached cover letter.

Reviewer 2 Report

The manuscript describes the identification of an intronic BRCA1 varina (likely to be pathogenetic) in ovarian cancer cases negative for BRCA1 and BRCA2 pathogenic mutations by applyinf their bioinformatic pipeline and filtering criteria on whole genome sequencing data from 22 cases of ovarian cancers from 22 familes, tested negative for BRCA1/2 pathogenic variants. Interestingly they identify a  variant of BRCA1 (c.5470-25T>A), not identified by commercial tests. By using SpliceAI, a relatively new in silico tool based on deep learning and pre-mRNA transcrip sequencing, they predicted a deleterious effect of this variant on BRCA1 trasncription. They then looked for the precense of this variant in other OC cohort. No claer to this referee how these ovariam cancer patients were selected. Could the authors explain?

However this variant was not new, as alreday reported by other different groups.

The informatic tool they used to detect the varian is able to capture cryptic splice sites located uo to 10 kilobase pairs from the exon-intron junctions.

The references are all mixzed up.

Even the variant they identified is not new, tha manuscript nicely rewieved the known intronic BRCA1 variants and discuss their clinical and therapeutic implications

Author Response

Please view the attached cover letter.

Round 2

Reviewer 1 Report

1) It is very important that the BRCA1 scientific community use the same nomenclature when describing variants to avoid confusion. As previously written all well-known databases use the BRCA1 transcript NM_007294.4. Moreover, this is the recommended transcript from the ENIGMA ACMG expert panel (guidelines soon to be approved by ClinGen). The alternative BRCA1 transcript NM_007300.4 encodes an 1884 amino acid protein. However, the authors show an 1863 amino acids protein in figure 2, which fits with the use of NM_007294.4. I strongly urge the authors to use the most common BRCA1 nomenclature.

2) In the results section (line 215-220), the authors mention that none of the prediction programmes predict that the c.5407-25T>A variants affects splicing. However, MaxEntScan predicts the generation of weak splice acceptor site when using Alamut (V2.15.0). Can the authors please explain the difference between Ensemble-based Variant Effect Predictor (VEP) and the use of MaxEntScan in Alamut?

3) Regarding the use of the wording de novo splice acceptor site I think it is a well-known phrase within the splicing community. However other words can explain this group of variants e.g. novel acceptor site. The use of the word cryptic is in my opinion simply wrong.

Author Response

Please find our response to the reviewer's comments in the attached cover letter.
